# Cellular Senescence: A Bridge Between Diabetes and Microangiopathy

**DOI:** 10.3390/biom14111361

**Published:** 2024-10-25

**Authors:** Jiahui Liu, Buyu Guo, Qianqian Liu, Guomao Zhu, Yaqi Wang, Na Wang, Yichen Yang, Songbo Fu

**Affiliations:** 1The First Clinical Medical College, Lanzhou University, Lanzhou 730000, China; jhliu21@lzu.edu.cn (J.L.); guoby21@lzu.edu.cn (B.G.); liuqq2020@lzu.edu.cn (Q.L.); zhugm18@lzu.edu.cn (G.Z.); wangyaqi19@lzu.edu.cn (Y.W.); wangna2021@lzu.edu.cn (N.W.); yangyichen430@126.com (Y.Y.); 2Department of Endocrinology, The First Hospital of Lanzhou University, Lanzhou 730000, China; 3Gansu Province Clinical Research Center for Endocrine Disease, Lanzhou 730000, China

**Keywords:** cellular senescence, diabetes mellitus, microvascular complications, diabetic retinopathy, diabetic nephropathy, diabetic cardiomyopathy

## Abstract

Cellular senescence is a state of permanent cell cycle arrest and plays an important role in many vascular lesions. This study found that the cells of diabetic patients have more characteristics of senescence, which may cause microvascular complications. Cell senescence, as one of the common fates of cells, links microangiopathy and diabetes. Cell senescence in a high-glucose environment can partially elucidate the mechanism of diabetic microangiopathy, and various types of cellular senescence induced by it can promote the progression of diabetic microangiopathy. Still, the molecular mechanism of microangiopathy-related cellular senescence has not yet been clearly studied. Building on recent research evidence, we herein summarize the fundamental mechanisms underlying the development of cellular senescence in various microangiopathies associated with diabetes. We gradually explain how cellular senescence serves as a key driver of diabetic microangiopathy. At the same time, the treatment of basic senescence mechanisms such as cellular senescence may have a great impact on the pathogenesis of the disease, may be more effective in preventing the development of diabetic microangiopathy, and may provide new ideas for the clinical treatment and prognosis of diabetic microangiopathy.

## 1. Introduction

Diabetes mellitus (DM) is a group of chronic metabolic diseases characterized by abnormal glucose metabolism due to defective insulin production and/or insulin action [1]. DM has become a global public health problem, and the number of people affected by it is growing rapidly. According to the United States Diabetes Association, the total economic burden of DM increased from USD 261 billion in 2012 to USD 327.3 billion in 2017, of which hospitalization due to DM vascular complications accounted for 43% of the total cost [2,3,4]. DM vascular complications are the leading cause of death in DM patients and are caused by the interaction of systemic metabolic abnormalities and local tissue responses caused by toxic metabolites [4]. Microvascular dysfunction is an important component of vascular complications in DM and may be a precursor to insulin resistance and T2DM, which can also exacerbate microvascular damage [5,6]. DM microvascular dysfunction may be related to endothelial dysfunction, smooth muscle cell metabolism disorders, and abnormal hormonal status, which further produce oxidative stress; alter cell signaling, gene expression, and protein function; and ultimately produce microvascular lesions [4,7].

Cellular senescence is a permanent state of cell cycle arrest, including replicative senescence (RS) and stress-induced premature senescence (SIPS) [8,9]. RS and SIPS are distinguished by the presence or absence of telomere changes [10], but at the same time, they share a common dependence on two major pathways to regulate cellular senescence: the p53/p21WAF1/CIP1 and p16INK4A/pRB pathways [11]. Senescent secretory phenotypes (SASPs) secreted by senescent cells include a range of pro-inflammatory cytokines and chemokines, which cause local or potentially systemic inflammation, disrupt tissue structure, and exacerbate cellular senescence and damage [12,13]. Cellular senescence involves a wide range of pathophysiological processes that may be involved in the development and prognosis of many diseases, especially age-related diseases [14]. Cellular senescence is known to be involved in many vascular diseases and plays an important role in many vascular lesions, such as atherosclerotic lesions, vascular smooth muscle cells in thrombosis, and the senescence of endothelial cells [15,16,17]. Recently, it was found that cellular senescence may also regulate the process of DM microangiopathy and may exacerbate oxidative stress in microvascular endothelial cells in high-glucose (HG) states [18], establishing pathogenic positive feedback loops that promote the development and accumulation of senescent cells in the organism and accelerate the pathophysiological process of DM [19,20]. However, the exact process has not been fully elucidated. Investigating the link between cellular senescence and related diseases may help to provide new research directions for the pathogenesis of DM microangiopathy.

In this review, we interconnect cellular senescence and diabetic microangiopathy and develop a picture of the underlying mechanisms of cellular senescence in DM microangiopathy. We conducted a literature review using keywords such as “cellular senescence”, “diabetes”, and “microvascular complications” in databases like PubMed and Scopus, focusing on studies published in the past decade. Drawing on emerging research evidence, this review explores how cellular senescence plays a critical role in diabetic microangiopathy, offering new insights for the clinical treatment and prognosis of the disease.

## 2. Cell Senescence Mechanism

Cellular senescence is a common cell fate that is considered to be a state of stress in which the cell cycle is forced to pause or even permanently stagnate due to a variety of stressors [8]. It is manifested by large and flat senescent cells [21], a significantly shortened telomere length [22], abnormal heterochromatin concentration [23], increased activity of negative regulators in cells such as p53/p21CIP1 and p16INK4a/pRb [24,25,26], the activation of β-galactosidase [27], and the production of aging characteristic secretions (e.g., SASPs) [8], among others. Currently, it has been found that the mechanism of cellular senescence may involve telomere and DNA damage, mitochondrial damage, endoplasmic reticulum stress, autophagy inhibition, inflammatory response, and oncogene dysbiosis (Figure 1).

### 2.1. Telomere and DNA Damage

Replicative senescence (RS) is the shortening of telomeres and reduction in proliferative potential resulting from multiple cell divisions in a cell, eventually leading to complete arrest. Eukaryotic DNA replication produces telomere deletions at the ends of the daughter strand [22]. When telomere deletion reaches a minimum threshold length, i.e., telomere unsealing, it fails to protect the DNA, leading to the opening of the DNA damage response (DDR), which activates the p53/p21 tumor-suppressor pathway and leads to cellular senescence [28]. After phosphorylated histone H2AX (γH2AX), which marks DNA breaks, accumulates at telomeres in human senescent cells [29], DDR signaling activates the key regulators ATM (Ataxia-Telangiectasia Mutated) and ATR (Ataxia Telangiectasia and Rad3-related) [30], and various proteins, including Chk1 and Chk2 (Checkpoint Kinases 1 and 2), respond to DNA damage with ATM- or ATR-dependent phosphorylation [30,31]. Subsequently, phosphorylated proteins activate numerous proteins, including p53, to slow the cell cycle. p53 regulates the complex antiproliferative transcriptional process [32,33], and it mainly induces the transcription of the cyclin-dependent kinase (CDK) inhibitor p21 in cell senescence and then blocks CDK2 activity, resulting in low phosphorylation of Rb (Retinoblastoma) and cell cycle arrest [24,34]. Therefore, the key to RS is the activation of DDR, which induces transient proliferative arrest, and when DNA damage cannot be repaired, DDR triggers cell senescence or apoptosis. Telomere shortening in leukocytes is associated with insulin resistance in diabetic patients, and telomere length not only has the potential to predict mortality in diabetic patients but can also be used as a good surrogate indicator for predicting diabetic complications [35,36], suggesting that RS is closely related to DM.

In response to stimulation induced by other stressors, DNA is similarly damaged and induces DDR activation, causing stress-induced premature senescence (SISP) [37,38]. For example, this can include endogenous by-products of cellular metabolism, reactive oxygen species (ROS), or reactive nitrogen species (RNS) [39], which can cause several types of DNA damage, including depurination/depyrimidine sites [40]. Exogenous potentially carcinogenic substances are also risk factors, such as ultraviolet rays [41], ionizing radiation (IR) of radioactive elements [42], chemotherapy drugs [43], nicotine, and biotoxins [44]. Without repair, these processes can lead to DNA mutations [45]. Thus, DDR allows cells to choose between DNA mutations and proliferative arrest, maintaining genome integrity and potentially triggering SISP [46].

### 2.2. Mitochondrial Damage

As mentioned earlier, most ROS from mitochondria in the body [47] accelerate replication aging [48]. ROS can activate p16^INK4a^/Rb [24,34,49] and the Ras-MAPK phosphate [50] signal channel, which eventually causes an upward increase in p21 [51]. Many ROS are also used to strengthen diabetic pathological pathways, leading to diabetic microvascular lesions [52,53,54]. The damage and dysfunction of the mitochondria itself can also induce cell senescence [8,55,56]. For example, HG may induce the aging of cortex cells by reducing the potential of mitochondrial membranes [57]. Mitochondrial dysfunction may cause the aging of pancreatic β cells to promote type II diabetes [58]. In addition, phosphorylated Rb is also detected by mitochondria levels, which may promote cell senescence by inhibiting E2F transcription factors to cope with various pressure stimuli such as DNA damage and hypoxia [59]. Although mitochondrial injury is widely considered a pathogenic factor in insulin resistance in DM, it is also an important pathway for cell senescence in DM. More mechanism research is still required to further describe the causal relationship between these two pathological phenomena, especially the interaction between the key regulatory components, such as electronic transmission chains [60], pRb [59], and the membrane potential [57].

### 2.3. Endogenic Reticulum Stress

Endoplasmic reticulum stress (ERS) is also one of the mechanisms of cell senescence. When an error occurs, or unpacking protein accumulates and exceeds the threshold, the ER will start one of the mechanisms of ERS, the unfolded protein response (UPR), to ensure the stable protein order in the intravascular mesh [61,62,63]. Although the signal pathways involved in UPR are mostly related to apoptosis, existing evidence has confirmed that activating transcription factor 6α (ATF6α), one of three ER transmembrane proteins involved in UPR signaling [64], mainly participates in cell senescence [65,66]. For example, the activation of ATF6α may be related to people’s fibroblast aging [67]. Another study found that after ultraviolet radiation, various ERS-related molecular signs, such as binding immunoglobulin protein (BiP) expression, were raised, which mediates cell senescence through the ATF6α axis [65]. In diabetic microvascular complications, ERS may be related to HG-induced glomerular and tubular damage [68]. In the premature aging of the renal tubular epithelial cells under HG, both the ATF4/P16 signaling pathway adjusted by ERS and endoplasmic reticulum stress-dependent p21 signals induced by the receptor for advanced glycation end-product (AGER) play an important role [69,70]. Therefore, improving tolerance to the internal quality network stress may become a new direction for targeted cell senescence therapy in diabetic microvascular lesions.

### 2.4. Cell Autophagy

Recent research and reports have confirmed a close relationship between senescence and autophagy [71]. Autophagy is regulated by genes; it includes the transport of intracellular components into lysosomal degradation, a stress response of cells to meet the needs of biological energy when resources are reduced [72]. Like aging, autophagy is a protection measure to help cells face various pressures [73]. At present, the research results provide different evidence for the role of autophagy in the senescence process: there may be an overlap between autophagy and senescence signaling pathways, but autophagy and aging are not mutually dependent. For example, in mitochondria, p53, which induces cell senescence, combines Parkin proteins in the PINK/PARK pathway induced by mitochondria to prevent them from entering the mitochondria and inhibiting autophagy in functional disorders [74]. The autophagy will produce a large amount of circulatory amino acids, which can be used to produce the SASP factor that promotes aging [75]. However, some researchers have found that autophagy and senescence can occur in parallel [76]. GATA4 gene mutations may promote the progress of type II diabetes [77], which may induce inflammation and aging by inhibiting the spontaneous autophagy of GATA4 [78]. In general, the choice between autophagy and senescence depends on the body’s reaction to the balancing disturbance, but the specific relationship between them has not yet been elucidated.

### 2.5. Inflammation Response

Senescence cells secrete the main component into a complex mixture of inflammatory factors, including tumor necrosis factor α (TNF-α), pantin (IL-1β, IL-2, and IL-6), and the trend factor (IL-8), called a “Senescence-associated secretory phenotype (SASP)”. These factors create cell microexis with mild inflammation, one of the main effects of senescent cells [8]. As inflammatory markers, IL-8 and IL-6 are specific components of SASPs and can also accelerate cell senescence [79,80,81]. IL-1 and IL-6 in SASPs can be used as a prognostic symbol of diseases such as T2DM, showing the harmful effect of inflammation [82]. Therefore, in aging, SASPs cause partial chronic inflammation, and they also accelerate cell senescence. A large amount of data indicate that the inflammatory markers such as (C-reactive protein) CRP in aging blood vessels are increased by TNF-α and IL-1β [83]; as mentioned earlier, inflammation can accelerate cell senescence and induce endothelial cell disorders, which may lead to the development of systemic microvascular lesions [84,85]. Some inflammatory factors in SASPs may aim to prevent or delay intervention measures related to aging-related vascular dysfunction.

### 2.6. Oncogene Disorders

Oncogene-induced senescence (OIS) is cell senescence caused by a cancer gene or cancer-suppressing gene disorders. The excessive proliferation caused by the expression of the primary cancer gene RAS [86] and the lack of cancer-suppression gene PTEN [87] affect the expression of p53 and suspend the cell cycle. OIS may be the middle ground for inducing cell senescence in different ways. For example, OIS is common in cell senescence caused by DNA injury and has played a role in cell senescence effects through cancer-suppressing genes Rb and p53 [25,88,89]. The OIS process contains the overlap of the inflammatory signal network; IL-8 and IL-6 are raised during OIS, mediating the OIS reaction. Other studies have found that chemokine receptor CXCR2 is involved in chemokine signaling in OIS [80,90]. In addition, OIS is also related to autophagy. In eternal cells, the expression of adenovirus cancer protein E1A inhibits RAS-induced aging and significantly reduces autophagy [91]. The inhibitory effect of autophagy delays OIS, which may include the acquisition of autophagy and its effects on the OIS phenomenon [92]. In the OIS process, the speed of glucose intake and utilization is slower [93], which may be related to the pathogenesis of DM. Therefore, OIS may interact with the signal of various cell senescence factors to play a role.

In recent years, studies have shown that changes in RNA levels [94], cellular metabolic reprogramming [95], and shifts in microbial communities [96] also play a role in the exit from the cell cycle and cellular senescence. This paper elucidates the molecular mechanisms linking cellular senescence to the relationship between diabetes and microangiopathy and proposes potential targeted therapeutic approaches to address these issues.

## 3. Cell Senescence—Vascular Cells in High-Glucose Environments

The microvascular system includes blood vessels between the first small arteries and the first small vein, consisting of endothelial cells (ECs), a small number of smooth muscle cells (SMCs), and pericyte and basal membranes (BMs) [97]. These components maintain the stability of blood flow dynamics and microcirculation in a normal state. However, in the HG environment, micro-blood vessels show obvious aging characteristics such as an increased surface area of vascular endothelial cells and SMCs and aging factors, such as increased acetyl-p53, p21, and SA-β-gal expression [98,99]. Multiple molecules play a key role in endothelial cell senescence under HG stress, mainly through oxidative stress. Among them, non-encoding RNA may be the hub regulating aging. DDR1 is a member of the recipient tyrosine kinase family that is elevated in the HG environment and participates in adjusting the expression of p53 and p21. Researchers have found that DDR1 is a target of miR-199a-3p, and miR-199a-3p/DDR1 path activation increases endothelial cell senescence [100]. Another microRNA, miR-200c, by lowering SIRT1 and eNOS mRNA, increases FOXO1 and p53 acetalization, reduces NO generation, and promotes aging and oxidation stress [101]. Protein disulfide isomerase (PDIS) is a radon oxidation enzyme [102] in the ER that is downgraded in diabetic mice. The knockdown of PDIA1 increases endothelial cell senescence. Further studies have found that the lack of PDIA1 can increase the subsulfonylation of Drp1 on Cys644, reduce Drp1 oxidation and restore state, increase mitochondria, and cause the dysfunction of endothelial cells in an HG condition [103]. The disorders of vascular inflammation may also cause cellular senescence under hyperglycemic conditions. High glucose levels alter key cellular pathways associated with growth and senescence, such as mTOR and NF-κB [104,105]. CTRP9 can reduce the phosphorylation of AMPK and the expression of KLF4 to relieve cell senescence and SASP secretion, control the inflammatory disorders of endothelial cells, and protect the good state of the vascular EC under HG stress [106]. CTRP9 is proven to maintain a normal lipid distribution and plaque stability [107], which can protect the EC and SMC through AMPK [107,108], indicating that CTRP9 may be a key lipid metabolism target. The inter-cytoplasm also participates in diabetic aging. KRAS is an oxidized target protein [109]. In the HG environment, the CircRNA-0077930 transmitted by the EC exterior body can pass the phagocytic effect to affect the CircRNA-0077930 of endogenous VSMC. It downregulates the expression of miR-622 and upregulates KRAS, which changes LDH and SOD in an unstable direction, damages the process of oxidative stress in VSMC, and eventually causes VSMC senescence [110]. Some scholars have found that the aging of pericytes may destroy the blood–brain barrier [111], but in the diabetic environment, the changes in aging pericytes have not been studied (Figure 2).

In addition to the cells that constitute the vascular structure, the aging of cells away from the blood vessels also affects blood vessel damage, such as the bone marrow endothelial progenitor cells (EPCs). Endothelial progenitor cells can be returned to the nest to the endothelium and mixed with neonatal endothelial cells [112], which is conducive to maintaining the health of blood vessels. Stem cell senescence and failure are considered important driving factors in the aging of the body [113]. Compared with the control group in one study, the telomerase activity of EPCs in the HG condition decreased, which may be due to stress-induced PGC-1α elevation in SIRT1 expression and positive feedback activation of p53 [114]. HG exposure creates a significant amount of SA-β-gal in macrophages and significantly increases IL-1β, IL-8, and IL-6 expression in SASPs, but the compensatory expression of antioxidant and anti-inflammatory genes SOD-1 and IL-10 decreases [115]. Neutral granulocytes can be treated by the cytokine components released by aging vascular units and then release neutrophil extracellular traps (NETs) at the pathological vascular generation site of the diabetic pathogen. NET enables neutrophils to clear aging ECs, which is conducive to reshaping the vascular system in the HG state [116]. These immune cells may be the factors involved in low-degree inflammation of diabetic microcirculation.

## 4. Cell Senescence—Microvascular Complications of Diabetes Mellitus

People with diabetes are more susceptible to age-related diseases, and their cells have more senescent features, such as shortened leukocyte telomeres [35]. This may result from stress-related aging, or in other words, diabetes itself may represent a state of aging. However, the cellular senescence caused by diabetic microangiopathy differs from physiological aging, and the former is mainly concentrated around the microvascular environment [117]. Vascular complications have a significant effect on the morbidity and mortality of DM [4]. Compared with large blood vessels, the microvascular structure is simpler, more widely distributed, and more permeable, and the influence of HG is more obvious. This causes various microvascular-related diabetic complications, such as diabetic retinopathy, diabetic nephropathy, and microvascular disease of the heart (Figure 3).

### 4.1. Diabetic Retinopathy

Diabetic retinopathy (DR) causes visual impairment in hundreds of millions of people worldwide [118]. DR is divided into non-proliferative DR and proliferative DR. In the non-proliferative DR period, the number of microaneurysms in small blood vessels increases; the proliferative DR period is characterized by the formation of new retinal blood vessels [119]. The mechanism that drives DR involves the multifactorial interaction of metabolic and hemodynamic factors [120]. Oxidative stress-induced aging is also the most critical step in the discovery of DR progression. For example, after a short period (48 h) of exposure of retinal pigment epithelium (RPE) to 25 mM glucose, the number of SA-β-Gal-positive cells increased in a time-dependent manner, and the cell cycle was stalled. Further studies found that HG increases H_2_O_2_, activates phosphorylation of p38, and upregulates intracellular Ca^2+^ levels, causing mitochondrial damage, activating the PI3K/AKT/mTOR pathway, increasing ROS levels in a positive cyclic manner in the case of fatty acid synthesis disorders, and promoting aging [121]. HG status not only promotes oxidative stress but also increases the effect of nitrification stress on retinal aging. ONOO-decomposition catalyst FeTPPS treatment can reverse the downregulation of SIRT1, p16^INK4A^, and miR34a in retinal tissue extracts of diabetic rats, reducing retinal aging [117]. In fact, in human retinal microvascular endothelial cells, all mitochondrial SIRT (SIRT3, 4, 5) mRNAs are significantly downregulated. Their upstream regulatory molecules miR-1 (target SIRT3), miR-19b (target SIRT4), and miR-320a (target SIRT5) are upregulated to varying degrees, and resveratrol can alleviate this disorder at mRNA levels [122] (Figure 4).

By analyzing the typical cell population present in the retina, Sergio et al. [98] found that Müller glial cells are richer in aging-related transcripts than cell types found in other retinopathies. The ultrastructure shows Müller cell organelle reduction, low phenotypic differentiation, and exacerbation of basement membrane (BM) material deposition [123]. This suggests that senescence in Müller cells in the HG condition may promote the progression of DR, but specific regulatory modalities have not been studied.

### 4.2. Diabetic Nephropathy

Diabetic nephropathy (DN) has become a major cause of chronic kidney disease (CKD) and kidney failure in developed countries, placing a huge burden on healthcare systems [124]. The occurrence of DN involves a variety of cellular senescence mechanisms, such as renal tubular epithelial cells (RTECs), glomerular mesangial cells (GMCs), podocytes, and endothelial cells (ECs) [124]. DM can induce various stresses in kidney cells to exacerbate aging, such as the accumulation of advanced glycosylation end products (AGEs), autophagy activation, oxidative stress, endoplasmic reticulum (ER) stress, mitochondrial damage, and inflammation [125] (Figure 5).

#### 4.2.1. Senescence of Renal Tubular Epithelial Cells (RTECs)

HG-induced aging of RTECs is an important cellular event that causes DN renal interstitial damage [126]. Mitochondrial regulatory factors PTEN induced putative kinase 1 (PINK1) and decreased optineurinase (OPTN) expression under HG stimulation, promoting mitochondrial damage by regulating mitochondrial autophagy and mtROS production and inducing RTEC senescence [126]. Regarding the increase in the number of senescent cells and the deposition of AGEs in the kidney tissues of patients with DN, research found that the expression of ER stress marker glucose regulatory protein 78 (GPR78), ATF4, and p16 in mouse RTECs cultured after AGE treatment was upregulated, promoting premature aging of RTECs [70]. Conversely, exogenous Klotho had a reversing effect on RTEC senescence, significantly attenuating age-induced Janus kinase 2 (JAK2) signal transduction and signal transducer and activator of transcription 1 and 3 (STAT1 and STAT3) [127]. Non-coding RNAs also play an important role in RTEC aging, such as miR-130a-3p having a negative regulatory effect on STAT3 through the MBNL1/miR-130a-3p/STAT3 pathway, thereby delaying the aging of mouse RTECs [128]. Another non-coding RNA, miR-378i, interacts with Skp2 to regulate PTEC aging in DN [20]. The above results suggest using non-coding RNAs in the future treatment and prediction of DN cell senescence and renal dysfunction. One study also found that HG can induce macrophages to secrete SASPs, forming a chronic inflammatory state in the kidneys [115].

#### 4.2.2. Senescence of Glomerular Mesangial Cell (GMCs)

GMCs play a role in the progression of senescence in DN cells through various processes, with non-coding RNA regulating the aging of GMCs in direct or indirect ways. MiR-126, as a protective factor, can delay HG-induced human glomerular mesangial cell (HGMC) senescence, and miR-126 upregulation can reduce p53, p21, and Rb expression in HGMCs through the telomere-p53-p21-Rb or Akt-p53-p21 signaling pathway, which is ultimately manifested by a shortened telomere length and cell cycle arrest [129,130]. miR-199a-5p or exogenous Klotho can also reduce inflammation and fibrosis of GMCs in HG-cultured rats by inhibiting the TLR4/NF-kB p65/NGAL signaling pathway, which can also inhibit the activity of the Ekr1/TLR4/mTOR axis and reduce inflammation and fibrosis of rat GMCs cultured in HG [131,132].

#### 4.2.3. Senescence of Podocytes

In recent years, podocyte senescence has been increasingly studied in DN, and podocyte damage is involved in the early progression of DN, manifesting as retraction, fusion, and disappearance, which may be related to oxidative stress and autophagy [133]. Nrf2 mediates endogenous antioxidant pathways, is a powerful anti-inflammatory and antioxidant, and promotes autophagy and other effects. One study found that in DN mouse models, β-hydroxybutyric acid could inhibit glomerular podocyte GSK3-enhanced Nrf2 activation, thereby alleviating podocyte aging and damage and improving diabetic glomerular lesions and proteinuria [134]. Klotho overexpression can induce the expression and activation of podocyte Nrf2 and its downstream target genes SOD2 and NQO1, inhibiting HG-induced oxidative stress and the apoptosis of podocytes [135]. Overexpressed Klotho also partially improves PKCα/p66shc-mediated podocyte damage and proteinuria [136].

#### 4.2.4. SASPs and Renal Cell Senescence

SASPs are associated with various cellular senescence in the kidneys and play a role in DN. In one study, SA-β-Gal was found increased in senescent PTECs, while the secretion of aging markers p16 and p21 increased, further promoting the secretion of SASPs and aggravated cellular senescence [70,137]. Conversely, CO can degrade SASPs by activating autophagy and improving cellular senescence in a variety of human and mouse kidney cells (RTECs, GMCs, and podocytes) in the case of DN [138], suggesting that SASP secretion may alter the outcome of DN.

Overall, these results suggest that hyperglycemia is a key driver of cellular senescence. RTECs occupy a central position in the glomeruli, and they not only control glomerular filtration but may also be involved in the response to local damage [139]. Most of the current research on DN cell senescence also focuses on RTECs, and relatively few other cell studies suggest that comprehensive studies of different cells are needed to reveal the mechanisms of cellular senescence in DN.

### 4.3. Microvascular Lesions of the Heart

T2DM is a risk factor for cardiovascular disease (CVD), increasing the risk of CVD death, and HG may weaken the ability of large and microvascular to protect against damage. Endothelial dysfunction of microvascular origin can lead to patients with non-obstructive coronary artery disease to develop microvascular angina at rest [140]. At present, there is less research on the central muscle aging regulation of diabetic cardiomyopathy (DCM). However, many key molecules that regulate aging have played an important role in the progression of DCM. SIRT3 can be used as a marker of aging in mice and can interact with nuclear layer proteins and heterochromatin-associated proteins to delay aging [141,142]. Similarly, in the myocardium of diabetic mice, SIRT3 expression decreased, NLRP3 inflammasomes were activated, and ROS accumulation increased, causing significant swelling of the mitochondria and disorders of myocardial structure and aggravating cardiac contractility and diastolic dysfunction in DM mice [143]. MiR-34a is highly expressed during the aging of the vasculature, induces VSMC and EC senescence and SASP secretion, and is regulated by SIRT1, Notch1, and B-cell lymphoma 2 [144]. miR-34a has also been reported to increase telomere wear and induce DNA damage by inhibiting PNUTS in aging hearts [145]. However, recent studies have found that in cardiomyocytes in the diabetic state, miR-34a has a more significant effect on regulating SIRT1 than PNUTS and causes changes in p53 expression levels [146]. This suggests that HG stress may differ from cardiomyocyte damage due to aging alone. P53 is highly expressed in diabetic cardiomyocytes and interacts with HIF-1α to promote MDM2-dependent ubiquitination and subsequent proteasome degradation in advanced DM, increasing cardiomyocyte senescence levels. At the same time, PFT-α can continuously inhibit p53 and reduce cardiomyocyte senescence [147]. Heart mesenchymal cells in diabetic patients also accelerate aging, slowing their growth rate and developing more senescent phenotypes, which may be associated with epigenetic enzyme disorders. Histone demethylase (jmjd3 and GNAT) acetylase GCN5 and p300/CBP-related factor (PCAF) in cardiac mesenchymal cells are reduced in HG. In contrast, the activation of PCAF with SPV106 can reduce SA-β-Gal accumulation to combat HG-induced senescence [148]. This suggests that epigenetics may be a more promising treatment for cellular senescence. However, HG-induced cardiomyocyte senescence is still in its infancy, and delaying cardiomyocyte senescence may improve prognosis levels in patients with DCM.

## 5. Treatment of DM Microvascular Cell Senescence Under HG Conditions

The accumulation of cellular senescence can lead to tissue dysfunction, exemplified by the fact that the accumulation of senescent endothelial cells in blood vessels affects the regeneration and angiogenesis capacity of the endothelium, impairs endothelial barrier function [149,150], and leads to vascular aging diseases. Therefore, selectively reducing the proportion of senescent cells in tissues may improve the prognosis of microvascular complications in patients with diabetes. At present, many drugs have been explored to delay cellular senescence in the diabetic state. Oxidative stress is the central link of cellular senescence in the HG state, and antioxidant treatment methods have important clinical prospects. Resveratrol is a strong antioxidant [151]. It has an anti-aging and anti-cancer effect [152,153], which can significantly reduce insulin resistance and glycosylated hemoglobin levels [154]; has therapeutic significance for restoring cellular homeostasis in diabetes; and has been shown to delay cellular senescence with time dependence in diabetic nephropathy and retinopathy [122,155]. Resveratrol is also considered an agonist of SIRT (an NAD+-dependent deacetylase). The consumption of SIRT leads to the senescence of endothelial cells in DM [122]. Resveratrol in DM can correct the expression state of SIRT by upregulating PPARδ activity [156]. For example, resveratrol increases the level of SIRT-1,2 in the T2DM myocardium and reduces the level of SIRT-3, but it is slightly different in T1DM and can reduce the abnormal morphology (hypertrophy/atrophy) of different diabetic cardiomyocytes and improve the myocardial contractility of the DM state [157]. SIRT1 also appears to be a key molecule for regulating HG stress and can be interconnected with various drugs in addition to resveratrol. For example, another antioxidant, ergothioneine, can upregulate SIRT1,6 and reduce the positive rate of endothelial cell SA-β-gal under HG [158]. Donaipizi [159] and Cudrania tricuspidata [160] also have a similar effect of activating SIRT1. In addition to delaying cellular senescence, targeting cells that eliminate senescence is also one of the strategies for treating cellular senescence, and in a recent study, dasatinib and quercetin were thought to selectively reduce the burden of aging on adipose tissue in diabetic kidney patients [161]. However, the therapeutic effect of targeting the elimination of senescent cells in diabetic microangiopathy has not been evaluated.

Improving or delaying cellular senescence may be one of the links where antiglycemic drugs improve microangiopathy. For example, metformin may delay HG-induced cellular senescence through various pathways. During the aging of the tubules in the HG condition, the expression of MBNL1 and miR-130a-3p is reduced. However, metformin can upregulate MBNL1; increase the stability of miR-130a-3p; and ultimately target STAT3, negatively regulate the expression of STAT3, and reduce the aging of renal tubules [128]. Metformin can also modulate the mTOR pathway by upregulating Klotho to reduce microvascular structural damage in diabetic nephropathy [162]. The hypoglycemic properties of antidiabetic drugs targeting SGLT-2 and GLP-1 targets have recently been found to be insufficient to explain their protective effects on the kidneys [163]. SGLT2 inhibitors such as dapagliflozin also exhibit non-hypoglycemic-dependent anti-aging effects, and β-HB elevation might promote the translocation of NRF2 from the cytoplasm to the nucleus, reducing oxidative stress [164]. Other studies have also found that β-HB can target the ATP binding site of GSK3b, promote Nrf2 activation, and reduce the aging of podocytes in the HG state [134]. Alogliptin can inhibit DDP-4 to increase the concentration of GLP-1, achieving a hypoglycemic effect [165]. Endoplasmic reticulum stress and cellular senescence of vascular endothelial cells have been reported to be relieved by alogliptin, possibly due to inhibition of ROS and inflammatory responses [166]. However, the anti-aging effects of hypoglycemic drugs in other microvascular lesions in diabetes have not been studied. Traditional Chinese medicine and its extracts have also been shown to relieve cellular senescence in DM microangiopathy, possibly achieved by reducing oxidative stress and chronic inflammation in different ways. This includes Shenkang injection [167], puerarin [133], Astragalus polysaccharides [168], ginseng, Panax notoginseng, and Sichuan root [169]. However, the therapeutic mechanism of other TCM physical stimulation therapies, such as acupuncture, has not yet been studied, and this may become a new research direction.

## 6. Conclusions

Diabetic microangiopathy is a common complication of diabetes mellitus, leading to a microvascular structure dysfunction with multiple manifestations, such as DR, DKD, and DCM. A comprehensive understanding of the mechanisms of diabetic microvascular disease is beneficial to improve the prognosis of patients with DM. However, the underlying molecular mechanisms associated with DM microvascular complications have not been fully studied. Nonetheless, cellular senescence in the HG environment can partially elucidate the mechanisms of diabetic microvascular disease. HG induces the aging of many types of cells and can promote the development of diabetic complications, including kidney disease, retinopathy, and cardiovascular disease. Most HG-induced aging is associated with oxidative stress, involving multiple molecular mechanisms such as SIRT and some oncogenes. These factors work synergistically to promote cellular senescence. However, the molecular mechanisms have not yet been fully studied, and the molecular activity evaluation criteria have not yet been established, which would clarify which molecules play a leading role in cellular senescence. The contribution of senescent cells to the causes of diabetic microvascular disease also needs to be explored.

Many reports have highlighted the important role of vascular endothelial cell senescence in producing ROS and pro-inflammatory responses. Still, endothelial cells do not play a role alone. The paracrine network of multiple cells in microvascular disease affects the lesion process during cellular senescence, so future research may need to focus on the intercellular communication of microvascular senescence in HG. Currently, most of the aging research in diabetic microvascular complications focuses on DKD, and there are fewer cellular senescence studies on DR and DCM. For example, the aging mechanism of Müller cells in DR has not been elucidated, which may become a new research direction.

In treating diabetic microangiopathy, delaying aging or eliminating senescent cells has application value. However, the role of drug therapy in different microangiopathies may differ, and their efficacy needs further evaluation. Furthermore, treating the mechanism of cellular senescence may change the existing diabetes treatment mode, but its clinical effect has yet to be evaluated. In conclusion, research on aging related to diabetic microangiopathy is still in its infancy; targeting the inhibition of cellular senescence provides a therapeutic strategy for diabetic microvascular disease, but more research is needed to elucidate the kinetics of cellular senescence formation in different cells under HG induction. This includes increasing our understanding of the molecular and cellular mechanisms underlying the pathophysiology of diabetic microvascular disease, designing effective strategies for preventing and treating microvascular lesions throughout the diabetic process, and conducting clinical trials based on interventions for cellular senescence.

## Figures and Tables

**Figure 1 biomolecules-14-01361-f001:**
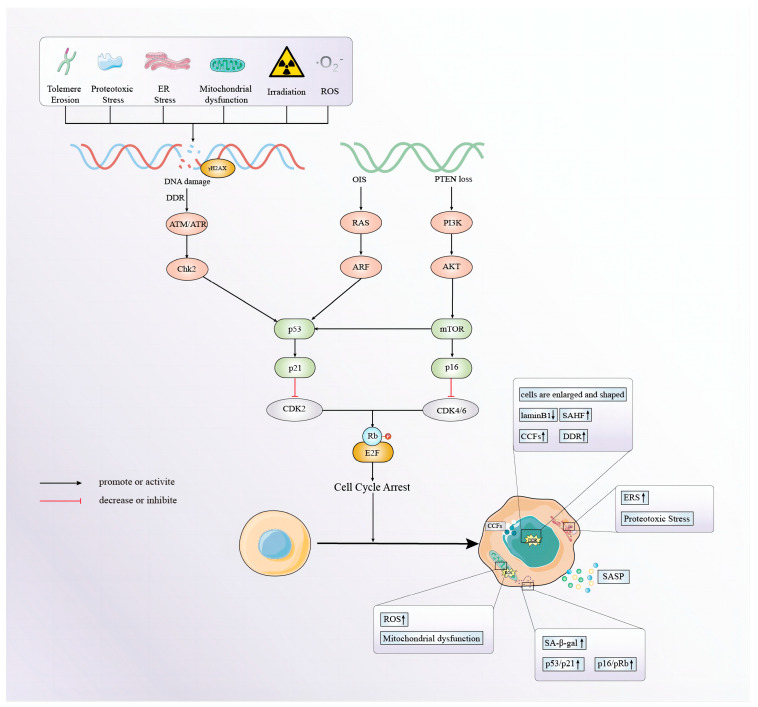
The main causes and mechanism of cellular senescence.

**Figure 2 biomolecules-14-01361-f002:**
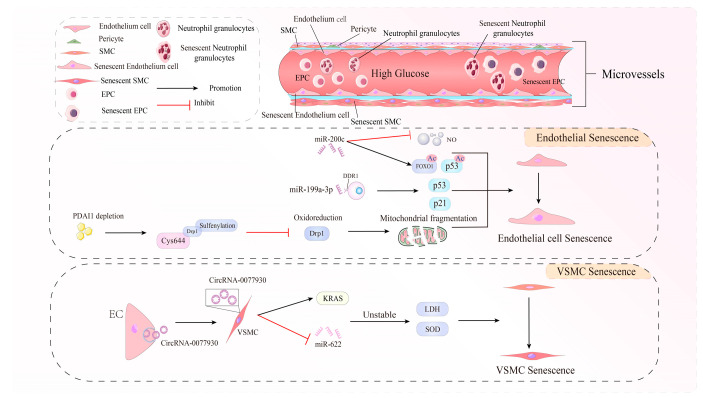
A schematic overview of how cellular senescence affects endothelial cells and smooth muscle cells and contributes to the pathogenesis of microangiopathy.

**Figure 3 biomolecules-14-01361-f003:**
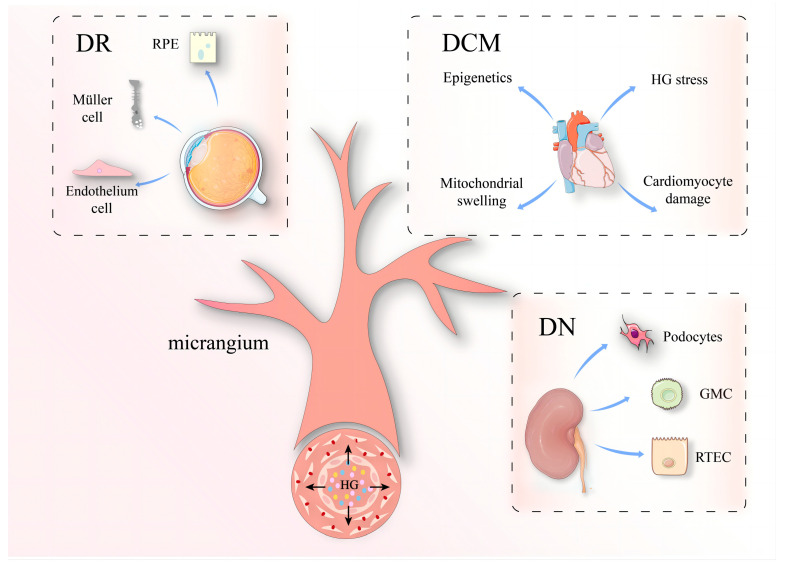
Complications of diabetic microangiopathy. HG can lead to a variety of microvascular-related diabetic complications, such as diabetic retinopathy (DR), diabetic nephropathy (DN), and diabetic cardiac micro-vasculopathy (DCM). RPE: retinal pigment epithelium; RTEC: renal tubular epithelial cell; GMC: glomerular mesangial cell.

**Figure 4 biomolecules-14-01361-f004:**
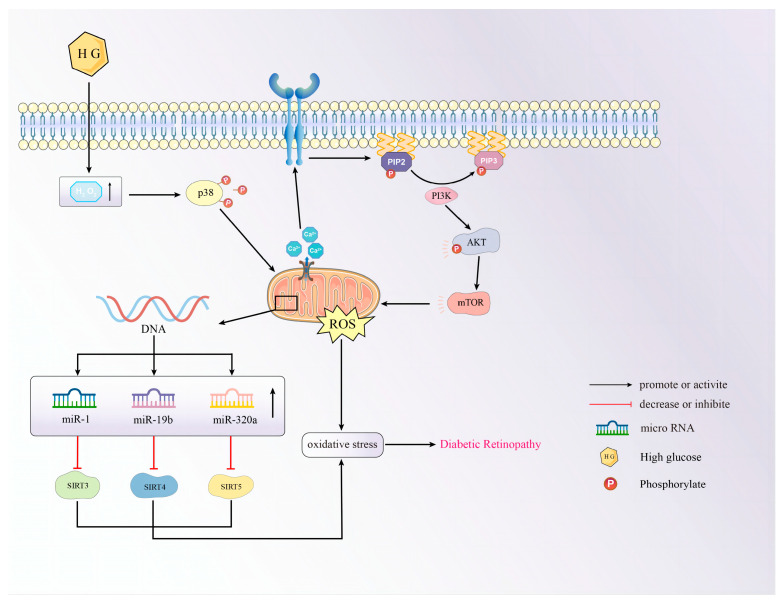
A schematic overview of how cellular senescence affects the pathogenesis of DR.

**Figure 5 biomolecules-14-01361-f005:**
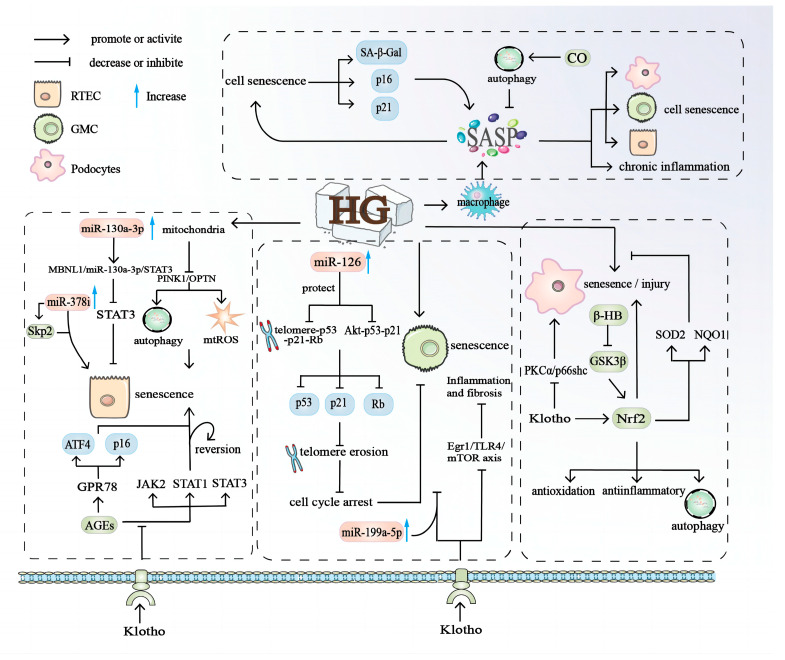
The aging mechanisms of RTECs, GMCs, podocytes, and SASPs play an important role in the cellular senescence of diabetic nephropathy. Mitochondria, AGE treatment, exogenous Klotho, and non-coding RNA-induced RTEC senescence in the HG environment. MiR-126, miR-199a-5p, and exogenous Klotho induced cell senescence in the HG environment. Nrf2, β-hydroxybutyric acid, and Klotho play a role in oxidative stress and the autophagy of podocytes in the early stage of DN under the HG environment. SASPs are associated with cell senescence in various renal cells (RTECs, GMCs, and podocytes). Cell senescence promotes β-galactosidase (SA-β-Gal) and senescence markers p16 and p21, promoting SASP secretion to further promote cell senescence. RTEC: renal tubular epithelial cell; GMC: glomerular mesangial cell; SASP: senescence-associated secretory phenotype.

## Data Availability

Not applicable.

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
