# Peer review of "Cellular Senescence: A Bridge Between Diabetes and Microangiopathy"

_biomolecules, 2024, doi:10.3390/biom14111361_

Round 1

Reviewer 1 Report

Comments and Suggestions for Authors

The authors tried to explain how cellular senescence is an important driver of diabetic microangiopathy and may provide new ideas for the clinical treatment and prognosis of DM microangiopathy. The aim is very interesting and potentially could lead to very important results or therapeutic improvement or suggestions. Unfortunately, the authors did not pay much attention to their text which is quite confusing and almost nonreadable. These are only some of the mistakes that should be corrected before continuing with the revision.

Please, revise the text and correct the repetitions.

Lines 48-49: Senescence secretion by senescent cells Senescent secretory phenotypes (SASPs) secreted by senescent cells include…. This does not mean anything. Please, correct

Lines 52-53: Cellular senescence involves a wide range of pathologies. Cellular senescence involves a wide range of pathophysiological processes that…. Same as above

Lines 54-56: Cellular senescence is known to be involved in many vascular diseases. Cellular senescence is known to play an important role in many vascular lesions, such as atherosclerotic lesions……again.

Lines 62-63: However, the exact process has not been fully elucidated. However, the exact process has not been fully elucidated

Lines 66-68: we interconnect cellular senescence and diabetic microangiopathy, 66 and develop a picture of the underlying mechanisms of cellular senescence and the study 67 of cellular senescence in DM microangiopathy…. Repetitions, again and again!!! This indicates a lack of attention in revising the text, which is unacceptable.

Figure 1. This is a figure. Please delete the phrase and add the explanations of the abbreviations used in the figure.

Line 85: Replicative Senescence (RS) is NOT the shortening of telomeres….But it is a consequence of telomere shortening

Line 86: multiple cell divisions in a cell    What does this mean?

Lines 109-111: For example, endogenous by-products of cellular metabolism reactive oxygen species (ROS) or reactive nitrogen species (RNS) [40], which can cause several types of DNA damage including depurination/depyrimidine sites.   What does this mean?  Something is missing.

Comments on the Quality of English Language

Many unclear phrases need revision. I could not proceed with my review after a few paragraphs.

Author Response

  1. Response to comment: The authors tried to explain how cellular senescence is an important driver of diabetic microangiopathy and may provide new ideas for the clinical treatment and prognosis of DM microangiopathy. The aim is very interesting and potentially could lead to very important results or therapeutic improvement or suggestions. Unfortunately, the authors did not pay much attention to their text which is quite confusing and almost nonreadable. These are only some of the mistakes that should be corrected before continuing with the revision.

Response: Thank you for your valuable comment on our manuscript. We have re-examined the entire text and utilized MDPI's author services for a thorough edit, which has helped us make the smoother.

Reviewer 2 Report

Comments and Suggestions for Authors

Please read the paper cautiously and fix the minute errors abundant throughout the text, like radon cells, return to the nest, missing words and so on. Eventually use a professional editing service.

Comments on the Quality of English Language

Probably translated using an automated translation service.  Please use a professional English editing service.

Author Response

  1. Response to comment: Please read the paper cautiously and fix the minute errors abundant throughout the text, like radon cells, return to the nest, missing words and so on. Eventually use a professional editing service.

              Response: Thank you for your valuable comment on our manuscript. We have revised the terminology and other questions you mentioned.We have re-examined the entire text and utilized MDPI's author services for a thorough edit, which has helped us make the smoother.

Reviewer 3 Report

Comments and Suggestions for Authors

A secretory phenotype associated with aging is acquired by cells as a result of cell aging, which aids in stopping cell division in response to stressful situations (oxidative stress, decreased nitrogen production, inflammation, etc.). It also affects chromatin organization and gene expression, and cells secrete pro-inflammatory cytokines, chemokines, growth factors, and proteases. Vascular problems are frequently the result of these alterations. Pathology of numerous organs and tissues is caused by metabolic and structural-functional changes in myocardial diabetes mellitus, increased oxidative stress, decreased microcirculation, and atherosclerosis, which are based on senescent changes in cells. 

One promising aspect of the paper is the authors' attempt to systematize vascular damage, diabetes mellitus, and senescent cell alterations in this review. Reviews have been conducted that emphasize the significance of cell senescence in the pathophysiology of diabetes and its different consequences, although PubMed indicates that there haven't been any of these over the previous five years. As an illustration: Narasimhan A, Flores RR, Robbins PD, Niedernhofer LJ. Role of Cellular Senescence in Type II Diabetes. Endocrinology. 2021 Oct 1;162(10):bqab136. doi: 10.1210/endocr/bqab136. PMID: 34363464; PMCID: PMC8386762.

Shen S, Ji C, Wei K. Cellular Senescence and Regulated Cell Death of Tubular Epithelial Cells in Diabetic Kidney Disease. Front Endocrinol (Lausanne). 2022 Jun 28;13:924299. doi: 10.3389/fendo.2022.924299. PMID: 35837297; PMCID: PMC9273736.

Lee JH, Lee J. Endoplasmic Reticulum (ER) Stress and Its Role in Pancreatic β-Cell Dysfunction and Senescence in Type 2 Diabetes. Int J Mol Sci. 2022 Apr 27;23(9):4843. doi: 10.3390/ijms23094843. PMID: 35563231; PMCID: PMC9104816.

The review is logically organized; general information about the different types of senescent changes in cells and the role of genes in this is provided first, followed by a discussion of their potential role in the pathophysiology of diabetes mellitus itself and vascular breakdowns that arise from cell senescence and damage inflicted by hyperglycemia. This is excellent work.

Such a review is pertinent given the absence of a consensus regarding the mechanisms underlying cell aging and, moreover, its significance in diabetes mellitus. There aren't any notable gaps in this field. A general understanding of the mechanisms of senescence and its role in the pathogenesis of diabetes mellitus and its vascular complications is provided by this review, which will be of interest to endocrinologists, cardiologists, nephrologists, optometrists, and doctors of a wide profile. The authors have also taken note of medications that can counteract adverse effects.

It should be mentioned that the authors regretfully only considered 30% of publications for a maximum of five years when composing the review. Apart from that, I have.Some of the pieces date back to before 2000. Indeed, these publications are added when needed, although connections to more recent ones may usually be discovered. No indications of excessive self-citation were present.

The authors reach a reasonable conclusion on the involvement of senescent cells in conjunction with hyperglycemia in vascular problems associated with diabetes mellitus based on the information provided.
It is possible to visually evaluate the points of application of an agent in the mechanism of development of vascular problems by using the relevant illustrative material that is available.

Among the work's drawbacks are the following:

1) a dearth of knowledge regarding the literature selection process (keywords, literature search engines, inclusion criteria for reviews, PRISMA, PICO);

2) there are numerous punctuation mistakes (lines 53, 75, 88, 89, 145, 194, 204, 208, 217, 234, 443, 448, 466-467); repeats (line 63); and other mistakes that the authors should thoroughly proofread and fix. The link to the drawings should ideally be before, not after, the period at the end of the sentence (lines 81, 242, 274, 299, 316); All drawings should have a name and comments attached to them; it is not necessary to specify that the drawing is a drawing in the title; it is recommended to include an abbreviation in the text (line 215); Avoid using outmoded terminology wherever possible (line 229);

3) If the writers permit repetitions within the talk, it would be preferable to integrate them in some way into a single block.

Author Response

  1. Response to comment: a dearth of knowledge regarding the literature selection process (keywords, literature search engines, inclusion criteria for reviews, PRISMA, PICO);

              Response: We sincerely appreciate your meaningful input on our manuscript. This article is a narrative review. While it doesn’t require the clear and standardized literature inclusion and screening criteria of a systematic review, we conducted searches using the keywords "cellular senescence," "diabetes," and "microvascular complications" in well-known databases like PubMed and Scopus before writing, ensuring comprehensive coverage of relevant literature. We focused on studies published in the past decade that directly discuss the mechanisms linking cellular senescence to diabetic microvascular complications. This was not previously mentioned in the text, possibly due to formatting, but we have now clarified this in lines 68-70.

  1. Response to comment:  there are numerous punctuation mistakes (lines 53, 75, 88, 89, 145, 194, 204, 208, 217, 234, 443, 448, 466-467); repeats (line 63); and other mistakes that the authors should thoroughly proofread and fix. The link to the drawings should ideally be before, not after, the period at the end of the sentence (lines 81, 242, 274, 299, 316); All drawings should have a name and comments attached to them; it is not necessary to specify that the drawing is a drawing in the title; it is recommended to include an abbreviation in the text (line 215); Avoid using outmoded terminology wherever possible (line 229);

              Response: Thank you for your valuable comment on our manuscript. We reviewed and corrected punctuation errors in the article. We also revised several abbreviations and updated outdated terminology. Additionally, we addressed issues with the figure captions.

  1. Response to comment:  If the writers permit repetitions within the talk, it would be preferable to integrate them in some way into a single block.

              Response: We are grateful for your constructive criticism of our manuscript. We've organized the various sections of the document and put it together.

  1. Response to comment:  It should be mentioned that the authors regretfully only considered 30% of publications for Apart from that, I have.Some of the pieces date back to before 2000. Indeed, these publications are added when needed, although connections to more recent ones may usually be discovered. a maximum of five years when composing the review.

              Response: Your valuable comments on our manuscript are greatly appreciated. We have added newer research perspectives and previous classic research perspectives and found commonalities between the two (e.g., lines 238-239, etc.).

Round 2

Reviewer 1 Report

Comments and Suggestions for Authors

The authors summarize the fundamental mechanisms underlying the development of cellular senescence in various microangiopathies associated with diabetes.

I appreciate the effort by the authors in revising the previous version of the manuscript. Now the paper is much easier to read, well written and I do not have many other comments to make, besides few corrections and clarifications needed (as indicated below).

Line 182: What do you mean with …inflammatory symbols in aging….?? Do you mean markers?? Please, clarify

Line 213 What are permeable cells??

Line 233 phosphoricization…. Please correct

Line 278 delete high glucose, it has been defined previously already 

Author Response

  1. Response to comment: Line 182: What do you mean with …inflammatory symbols in aging….?? Do you mean markers?? Please, clarify f the mistakes that should be corrected before continuing with the revision.

Response: Thank you for your valuable comment on our manuscript. We indeed want to express the concept of "inflammatory markers such as CRP", and we have now corrected it to "inflammatory markers".

  1. Response to comment: What are permeable cells??

              Response: We initially used the term 'permeable cells' mistakenly when we intended to refer to 'pericytes.' We have corrected this in the manuscript to accurately reflect our intention. Thank you for bringing this to our attention.

  1. Response to comment: phosphoricization…. Please correct

              Response: Thank you for your comment regarding 'phosphoricization.' We have corrected this term to 'phosphorylation' in the revised manuscript. We appreciate your attention to this detail.

  1. Response to comment:  delete high glucose, it has been defined previously already 

              Response: Thank you for your suggestion. We have removed the mention of 'high glucose' as it has been defined previously in the manuscript. We appreciate your input.